# Vaginal Capsules: A Viable Alternative for the Delivery of *Lactobacillus* spp.

**DOI:** 10.3390/microorganisms13051056

**Published:** 2025-05-01

**Authors:** Leandra Sá de Lima, Lívia Custódio Pereira, Rosane Mansan Almeida, Yanna Karla de Medeiros Nóbrega

**Affiliations:** 1Clinical Microbiology and Immunology Laboratory, Department of Pharmacy, University of Brasilia, Campus Darcy Ribeiro, Brasilia 70910-900, DF, Brazil; leandra.farmacotecnica@gmail.com (L.S.d.L.); liviacp78@gmail.com (L.C.P.); rmansan@gmail.com (R.M.A.); 2Vulvar Pathology Clinic, Department of Gynecology, University Hospital of Brasilia, University of Brasilia, Brasilia 70910-900, DF, Brazil

**Keywords:** *Lactobacillus* spp., vaginal capsules, probiotics, magistral formulations, vaginal infections treatment

## Abstract

*Lactobacillus* spp. and other beneficial bacteria are predominant in the vaginal microbiota and represent an opportunity to correct dysbiosis if administered intravaginally. Since no commercial formulations are available, developing magistral formulations is an option, provided that they ensure viability and therapeutic efficacy. To evaluate their stability and culturability, four magistral formulations containing 10^9^ microorganisms were tested: vaginal capsules, vaginal ointment, gelatinous ovules, and waxy ovules. Certified strains of *L. crispatus*, *L. johnsonii*, *L. gasseri*, *Limosilactobacillus reuteri*, and *Lacticaseibacillus rhamnosus*, as well as a combination of the five, were used. The formulations were tested for pharmaco-technical stability using average weight and disintegration tests, as well as evaluation organoleptic. In addition, microbial recovery was evaluated by counting Colony-Forming Units (CFUs). All forms, except the gelatinous ovules, allowed microbial recovery at concentrations from 10^7^ to 10^9^ CFUs, ensuring stability for 60 days. The recovery varied depending on the strains and dosage forms employed, with the most favorable outcomes for vaginal capsules. This highlights the need for standardized strains and excipients in magistral formulations. Further studies are needed to evaluate the viability of other strains of different excipients, vehicles, or different storage; however, capsules have demonstrated efficacy and are an excellent candidate for vaginal use formulations of *Lactobacillus* spp.

## 1. Introduction

The human vaginal microbiota is predominantly composed of species from the genus *Lactobacillus*, some of which were recently reclassified into different genera [1]. It encompasses numerous biological functions, such as protection against infections, including sexually transmitted diseases, protection against HPV-induced neoplasia [2,3], and maintenance of the necessary balance for reproductive function [4].

Ravel et al. [5] studied the vaginal microbiota and proposed a classification into Community State Types (CSTs), which is currently accepted. CST I, II, and V correspond to healthy microbiota and are dominated by *L. crispatus*, *L. gasseri*, and *L. jensenii*, respectively. CST IV A and CST IV B do not show a significant presence of *Lactobacillus* species. Although they are often found in asymptomatic women, they represent a state called dysbiosis, as they may promote the development of infections [5]. CST III, characterized by the dominance of *L. iners*, could represent a state of “evolving dysbiosis” due to its lower lactic acid production and the controversial nature of its protective role [6].

As our knowledge of the vaginal microbiota increases, so does the use of *Lactobacillus* spp. and other probiotic bacteria probiotics as adjuncts in the treatment or prevention of vaginal infections. Among the possible routes of administration, intravaginal administration is quite favorable for correcting the vaginal microbiota [7,8]. However, there are no commercially available formulations in many countries, and the product must be obtained by compounding magistral formulations for which there is no specific standardization.

Hill (2014) [9] stated that for a pharmaceutical product to be classified as a probiotic, the strain must be isolated, thoroughly characterized, viable, and exhibit health benefits. Several species have been studied to treat bacterial vaginosis, including *Lacticaseibacillus rhamnosus*, *Lacticaseibacillus reuteri*, *L. plantarum*, and *L. acidophilus* [10]. The number of microorganisms used also varies. Doses ranging from 10 million (10^7^) to 10 billion (10^10^) Colony-Forming Units (CFU) of exogenous *Lactobacillus* spp. administered orally or vaginally have shown greater efficacy, suggesting that the amount used is essential to the effectiveness of the formulations [11].

The viability of microorganisms can be assessed essentially by three aspects: reproducibility, cell/membrane integrity, and metabolic activity. Although many bacteria can exist in a viable but non-culturable state, reproducibility and recovery as CFU are well-accepted parameters to demonstrate viability [12].

Considering the potential benefits of probiotics and the need for standardization, this study aimed to evaluate the stability and viability of different probiotic strains and to quantify their recovery percentage from four pharmaceutical forms for vaginal application: vaginal capsules, vaginal ointment, gelatinous ovules, and waxy ovules. Strains of species that occur naturally in the vaginal or intestinal microbiota were selected, including *L. crispatus*, *Limosilactobacillus reuteri*, *L. johnsonii*, *L. gasseri*, and *Lacticaseibacillus rhamnosus*, along with a composition comprising the five (L5 formula). The concentration used for each strain was 1 billion (10^9^) CFU.

## 2. Materials and Methods

### 2.1. Origin and Maintenance of Strains

Five strains of *Lactobacillus* spp. were selected (Table 1), kindly provided by Farmacotécnica Instituto de Manipulações Farmacêuticas Ltda (Farmacotécnica). The research was performed in vitro using commercial and certified strains, and ethical approval was not required. *L. crispatus* and *L. gasseri* were selected because of their dominance in the healthy vaginal microbiota [2]. In addition, *Limosilactobacillus reuteri*, *L. johnsonii*, and *Lacticaseibacillus rhamnosus* are species that have been extensively studied in various types of research, including clinical trials [13,14,15].

Bacteria were cultured in MRS Broth Acumedia^®^ (Neogen do Brasil) medium, which consisted of 1% enzymatic digest of animal tissues, 1% meat extract, 0.5% yeast extract, 2% dextrose, 0.5% sodium acetate, 0.1% polysorbate 80, 0.2% disodium phosphate, 0.2% ammonium citrate, 0.01% magnesium sulfate, and 0.005% manganese sulfate; pH was adjusted to 6.5.

### 2.2. Preparation of Formulas

The formulations were individually prepared at Farmacotécnica Inst. de Man. Farmacêuticas Ltda, Brasília, Distrito Federal, Brazil. Preparation techniques followed regulatory and technical requirements. *Lactobacillus* spp. were incorporated into four pharmaceutical forms: vaginal capsules, vaginal ointment, gelatinous ovules, and waxy ovules. Gelatinous ovules have hydrophilic properties, while waxy ovules are lipophilic. Each strain was incorporated separately into all four presentations at a concentration of 10^9^ CFUs per unit of dosage form and a combination of one billion CFUs for each strain (L5 formula), for a total of 5 × 10^9^ CFUs per unit (Table 2).

To prepare the formulations, each ingredient was weighed on a certified precision balance, and the powders were sieved before homogenization. Weighing calculations for the Active Pharmaceutical Ingredients (API) were performed automatically by the *Fórmula Certa* computerized system (Fragontech, Jundiaí, São Paulo, Brazil), which considers the weight, density of the powders and apparent volume to suggest the weight of the excipient to be used. The capsules were then prepared using a semi-automatic encapsulation machine at room temperature (~25 °C). The capsules were commercially obtained and consisted of a water-soluble polysaccharide derived from *Manioht esculenta* (cassava) starch through a natural fermentation process. These capsules exhibit rapid disintegration when in contact with aqueous environments.

Lactose was chosen as an excipient for the capsules, as it is a simple carbohydrate derived from milk, with a low cost and good drug flow, release, and bioavailability characteristics [16]. Its applicability in vaginal capsules has been studied due to the possibility of favoring the stability of probiotics in solid formulations [17,18].

Because of its hygroscopic properties, lactose acts as a plasticity agent during water absorption, forming stable amorphous crystalline structures [19]. It is an excellent protective agent during the dehydration of probiotics by the spray drying method, as its hydroxyl group interacts with phosphate groups present on the surfaces of the lipid bilayers of microbial cells, replacing the hydrogen bonds broken by the removal of water from the environment [20].

Petrolatum (CAS: 8009-03-8) was chosen as an excipient for preparing vaginal ointments. The choice was aimed at maintaining a completely anhydrous galenic base, which is expected to favor the viability of the strains with little intervention, avoiding a high number of excipients that could interfere with the results. Petrolatum has a long history of use in dermatology, dating back to the 1800s with safe application [21,22]. After weighing, the *Lactobacillus* spp. were cold-conveyed with homogenization in a mortar and pestle.

The excipient for waxy vaginal ovules was prepared using *Appis mellifera* Linnaeus (beeswax) and *Theobroma cacao* butter (cocoa butter). The disadvantage of this preparation is the need to keep it refrigerated to avoid melting of the formula caused by the high ambient temperatures in Brazil. Waxy ovules were prepared by weighing and sieving the *Lactobacillus* spp. to ensure particle uniformity. The components of the ovule mass were weighed and transferred to a sanitary stainless-steel container and placed in a bain-marie for complete melting under controlled temperature, which should not exceed 60 °C, and mixed until complete homogenization.

After the excipient was completely melted, the probiotics were added and homogenized. Still in liquid form, they were packaged in disposable polypropylene ovule molds under individual weighing. The molds were refrigerated and, after complete solidification, sealed in an automatic sealer.

The excipient of the gelatin vaginal ovule was prepared using disodium ethylenediaminetetraacetic acid (EDTA), vegetable glycerin, powdered gelatin, potassium sorbate, and water. This excipient was chosen to test the stability of the different strains in the presence of water. It is a formulation with a homogeneous appearance that is stable at room temperature.

The components of the ovules were weighed and transferred to a sanitary stainless-steel container and placed in a water bath under controlled temperature (50 °C to 55 °C) until complete melting and mixed until complete homogenization. The *Lactobacillus* spp. were weighed and sieved to ensure particle uniformity. After the complete melting of the excipient, the probiotics were added and homogenized. Still in liquid form, they were packaged in disposable polypropylene molds for ovules, individually weighed, and, after complete solidification, sealed in an automatic sealer.

All tests were conducted at zero (0), 30, and 60 days to evaluate the stability and recoverability of *Lactobacillus* spp. Two different experiments were carried out, each of them in duplicate. The table below summarizes the composition of all tested formulations.

### 2.3. Pharmaceutical Analysis

#### 2.3.1. Average Weight Analysis

After preparation, each formulation was analyzed for average weight using the Farma SP 5000 statistical processor (Gehaka, São Paulo, Brazil). The method used to evaluate the average weight of magistral capsules was described in the National Formulary of the Brazilian Pharmacopoeia [23], in which intact capsules were individually weighed, and the average weight was determined in grams. The formula was considered to comply if none of the capsules fell outside the range of ±10% (up to 300 mg) or ±7.5% (above 300 mg) [23].

#### 2.3.2. Disintegration Test

The procedure was adapted from the Brazilian Pharmacopoeia, 6th edition [14]. A water bath at 37 °C was used. Each dosage form was added to an Erlenmeyer flask containing 100 mL of 37 °C water and stirred in the water bath while the dissolution time was recorded. Three units of each dosage form were evaluated. Complete disintegration was considered to have occurred when the capsule or ovoid showed: (a) complete dissolution; (b) a complete separation of its components, with melted fatty substances accumulating on the surface or insoluble powders settling to the bottom of the container, while soluble components dissolved; or (c) rupture of the gelatinous capsule, allowing the release of its components. The disintegration time should not exceed 30 min for any of the dosage forms tested [24].

#### 2.3.3. Organoleptic Analysis

Organoleptic evaluations were performed for each formulation, observing the appearance, uniformity, and absence of lumps. Unfortunately, anhydrous vehicles did not allow pH testing or other types of evaluation.

### 2.4. Viability Testing of Strains of Lactobacillus *spp.*

A preliminary culture was performed to assess the viability of the strains of *Lactobacillus* spp. used. We tested tubes containing 2 mL MRS broth by adding 10 µg of each lyophilized strains of *Lactobacillus* spp., which equates to roughly 500,000 CFU/mL per the manufacturer’s instructions. After incubation at 37 °C (±2 °C) for 12 h in a water bath, 10 µL of each culture was seeded onto MRS agar and cultured for 48 h at 37 °C with 5% CO^2^ and 95% moisture or in an anaerobic flask under the same conditions.

### 2.5. Culturability Assay of Recovered Strains of Lactobacillus *spp.* from Vaginal Preparations

Sterile tubes containing 30 mL of MRS broth were inoculated with one unit of each formulation: 3 g vaginal cream, a gelatinous ovule, a waxy ovule (both ~3 g), and a vaginal capsule (~500 mg) [13]. The concentration of 10^9^ CFU was maintained for all formulations except for the combination formula, which used 10^9^ CFU of each strain for 5 × 10^9^ CFU, and the vaginal cream (10^9^ CFU/gram). These amounts were used for weight equivalency with vaginal ovule and to approximate the treatment dosage used. Tubes containing each formula without microorganisms were used as negative controls.

The tubes were incubated in a water bath at 37 °C (±2 °C) for 12 h to simulate the prescribed vaginal use time (overnight). At the end of this period, two 1:100 dilutions (final dilution 1:10,000) were prepared. For CFU enumeration, 10 µL were plated on MRS agar plates, incubated at 37 °C with 5% CO_2_ and 95% moisture, and in an anaerobic vessel under the same conditions for 48 h.

### 2.6. Data Analysis

The CFU data were analyzed using Microsoft Office Excel software 2021 (Microsoft). The figures were constructed using GraphPad Prism version 10.0 software for Windows (La Jolla, CA, USA) and represent the average percentage from two independent experiments with biological duplicates.

## 3. Results

### 3.1. Pharmacotechnical Evaluation of Vaginal Formulations

#### 3.1.1. Average Weight

The average weight was estimated for the capsules. According to the Brazilian Pharmacopeia [24], the limit for the average weight of manipulated capsules above 300 mg can vary up to 7.5%. Considering that the manipulated capsules had an average weight of approximately 0.5 g or 500 mg, the acceptable variation represents about 37.5 mg, either added or subtracted from the final weight. The variation observed between capsules of different strains is likely due to the variability in powder density and lyophilized strains of *Lactobacillus* spp. The average weight of the vaginal capsules remained within the required parameters and complied with them (Table 3).

#### 3.1.2. Disintegration Tests

The average dissolution time of the capsules and vaginal ovules was evaluated. For the petrolatum-based vaginal ointment, dissolution measurement was not possible. The average disintegration time of the vaginal capsules was less than five minutes for all *Lactobacillus* spp. capsules, including the L5 formula (Table 4). During the procedure, a whitish sediment was observed that slightly clouded the solution during homogenization. The capsule shell also wholly disintegrated during the observation period.

The average disintegration time of the waxy ovules was less than 12 min (Table 4). The dissolution of the waxy ovules met pharmacopeial specifications, although with a longer dissolution time than that observed for the vaginal capsules. During the procedure, complete fusion of the formulation was observed with the formation of an oily supernatant in each Erlenmeyer flask, which did not mix with water after complete disintegration.

The disintegration of the gelatinous ovules was observed to begin at six minutes and did not exceed 24 min (Table 4). The average time meets the requirements of the National Formulary, which ensures compliance, but it represents the longest dissolution time among the studied forms. During the procedure, complete fusion of the ovules was not observed, as it maintained a gelatinous consistency, although more fluid. In addition, no formation of supernatants or clouding of the medium was observed.

#### 3.1.3. Organoleptic Evaluation of the Vaginal Ointment

The evaluation of the organoleptic characteristics showed a uniform visual distribution, without granules perceptible to the naked eye and impalpable on spatulation. All samples analyzed had a semi-solid texture and milky-white color with a greasy characteristic. There are no other applicable analyses for this type of composition.

### 3.2. Microbiological Evaluation of Vaginal Formulations

#### 3.2.1. Recovery of *Lactobacillus* spp. from Vaginal Capsules

All strains of *Lactobacillus* spp. could be recovered from the vaginal capsules. Overall, we observed that the *Lacticaseibacillus rhamnosus* strain had a higher recovery capacity, exceeding 100%, and stability over time in this formulation. On the other hand, the *L. johnsonii* strain showed a lower recovery percentage, below 50%, at all-time points studied. *Lactobacillus gasseri* was also found to recover less than 50% at time zero. The remaining strains of *Lactobacillus* studied showed a recovery percentage equal to or greater than 50% (Figure 1). A recovery rate greater than 100% indicates microbial growth.

#### 3.2.2. Recovery of *Lactobacillus* spp. from Vaginal Ointment

All strains of *Lactobacillus* spp. could be recovered, both individually and in the L5 formula, when formulated in the vaginal ointment. *L. crispatus*, *Lacticaseibacillus reuteri*, *Lacticaseibacillus rhamnosus*, and L5 formula showed higher growth rates, while *L. johnsonii* and *L. gasseri* showed lower growth rates at all time points studied (Figure 2).

*L. johnsonii* and *L. gasseri* had a low recovery rate, less than 20%, with *L. gasseri* showing more excellent stability over time (Figure 2). *Lacticaseibacillus reuteri*, *Lacticaseibacillus rhamnosus*, and the L5 formula had higher recovery rates. Still, all strains had recovery rates below 50%, except for *Lacticaseibacillus reuteri* (at 30 days) and L5 formula (at time zero), which sporadically had recovery rates close to 80% (Figure 2).

#### 3.2.3. Recovery of *Lactobacillus* spp. from Waxy Ovules

The waxy ovules provided varying degrees of recovery for all encapsulated strains at all time points evaluated, except for *L. crispatus*. At time zero, only strain L5 showed a recovery percentage above 100%, with a linear decrease in this percentage at subsequent time points (Figure 3). *L. crispatus* showed a low recovery percentage at 30 days and was not recovered at 60 days. At 30 days, *L. johnsonii*, *L. gasseri*, *Lacticaseibacillus reuteri*, *Lacticaseibacillus rhamnosus*, and the L5 formula showed recovery percentages ranging from 35% to 150%. *L. gasseri* and *Lacticaseibacillus reuteri* achieved percentages greater than 100% at 60 days. In contrast, the L5 formula showed a decrease in strain recovery over time (Figure 3). The behavior of this dosage form showed significant variability among different strains at all three time points, with no evidence of linearity or proportionality.

#### 3.2.4. Recovery of *Lactobacillus* spp. from Gelatinous Ovules

The gelatinous ovules were unsuitable for encapsulating the studied strains, as no strain recovery was observed at any time points evaluated.

### 3.3. Potential Validity of the Dosage Forms Studied

Current regulatory requirements state that the declared concentration should be detectable at any time within the stated shelf life. Although the initial concentration of one billion was recovered only for the *Lacticaseibacillus rhamnosus* vaginal capsules, lower concentrations could be declared for the other dosage forms. Table 5 summarizes the highest concentration that could be claimed for each microorganism in different formulations.

Throughout the present study, only formulations that allowed recovery between 10^7^ and 10^9^ were considered viable, which, according to the literature, is the ideal dose to use in humans [25,26,27].

*L. crispatus* encapsulated in waxy ovules and *L. gasseri* encapsulated in vaginal capsules remained stable for 30 and 60 days at a concentration of 10^7^ CFU. However, all other formulations remained in the range of 10^8^ CFU per dosage unit at the 60-day mark. Therefore, encapsulation of *L. johnsonii* in vaginal cream was not considered viable.

## 4. Discussion

Probiotic strains of *Lactobacillus* spp. have proven to be great allies in the fight against vaginal infections, as they exhibit microbicidal activity against difficult-to-control pathogens, such as antifungal-resistant strains of *Candida* spp. [28]. However, for a probiotic microorganism to promote health benefits, it must be administered sufficiently and actively at the intended application site [9]. Therefore, *Lactobacillus* spp. must withstand the technological processes required to produce pharmaceutical forms suitable for the intended use and maintain their viability throughout the established shelf life.

Choosing the most convenient dosage form favors patient compliance [29]. However, there may be more appropriate forms to ensure the survival and subsequent activity of the probiotic.

In the present study, four dosage forms were selected for use with intravaginal probiotics: hard gelatin vaginal capsules, petrolatum-based vaginal cream, waxy vaginal ovules, and gelatinous vaginal ovules. The gelatin ovules, although a convenient application, did not allow for the recovery of any strains, making them unsuitable for encapsulation.

The excipient in the gelatinous ovule formulation may have contributed to this failure, as it contained 43.0% glycerol. Camilletti et al. performed an in vitro test with different glycerol levels to evaluate the viability of *L. fermentum* and *Lacticaseibacillus rhamnosus* strains [13]. They found that strain viability decreased with glycerol levels above 30%. The presence of sorbic acid in the formulation may also have affected the viability of the strains. Georgieva et al. demonstrated that adding potassium sorbate (0.5 to 1.0%) reduces the growth capacity of *Lactobacillus plantarum* [30]. However, the gelatin used likely hindered the release of probiotics. Different gelatins have different viscosities and melting points [31], and the gelatin used in the production of the ovules did not wholly dissolve at the temperature used in the tests.

When evaluating the vaginal ointments with petrolatum as the oily vehicle, there was a 1-log reduction in CFU recovered at 60 days for most strains, except for *L. johnsonii*, which was not adequately retrieved at any time point. Puebla-Barragan et al. [22] studied the viability of a commercial formulation of *Lacticaseibacillus rhamnosus* (LGR-1) and *Lacticaseibacillus reuteri* (LRC-14) in different vehicles. They found a viability loss of approximately 59.3% in oily vehicles, with coconut oil and vaseline showing the lowest losses. In the current study, a similar loss of viability is observed at the 30-day mark. However, at 60 days, the loss of viability was significantly higher than in the Puebla-Barragan study, possibly due to the use of different strains of *Lactobacillus*.

All strains could be recovered regarding waxy ovules, although some, such as *L. crispatus*, showed low recovery capacity. In the development of vaginal ovules, Camiletti et al. (2018) observed a reduction of 1 to 2 logs in the different compositions studied [13]. The fusion and shaping of vaginal ovules involve controlled heating, which, although not exceeding 60 °C, may have affected the viability of the strains. Among the processes used for pharmaceutical forms, the heating process has the most potential for degradation of the microorganisms used. In a study evaluating the growth of the strains of *Lactobacillus* spp. exposed to different temperatures (37 °C, 42 °C, 50 °C, and 55 °C) for 24 h and re-cultivated, Silva et al. (2020) observed that strains exposed to temperatures above 50 °C did not retain their growth capacity [32].

Water activity is a parameter that measures water availability and is inversely related to probiotic viability after prolonged storage. It is generally observed that viability decreases as water activity increases [33]. One strategy to reduce water content is to use lipophilic excipients, which can form an emollient film that is comfortable for vaginal application. Both the vaginal cream and the waxy beads retained the lipophilic property. Both formulations allowed CFU recovery, with the cream showing lower strain viability rates than the ovules and lower than the vaginal capsules.

In vaginal capsules, most strains can be recovered in higher proportions than other forms, suggesting that this pharmaceutical form may lead to increased vaginal colonization by *Lactobacillus* spp. These findings are consistent with a clinical trial evaluating the safety of vaginally administered gelatin capsules containing probiotics in healthy, sexually active women [34]. The main results obtained were a reduction in the Nugent score and an increase in the number of cultivable *Lactobacillus* spp. The capsules containing *Lactobacillus* spp. were well tolerated by the patients and considered an alternative for restoring the vaginal microbiota.

Capsules also represent a dry presentation that improves the viability of *Lactobacillus* spp. The excipient used was lactose, chosen for its potential lyoprotective effect and possibly as a nutrient for *Lactobacillus* spp. when released in the medium [35]. Although capsules are not the most common presentation for vaginal use, their excellent performance with rapid dissolution and higher recovery of *Lactobacillus* spp. makes them the best candidate among the dosage forms studied here for these probiotics.

As important as the dosage form is the dosage used. Although there is no consensus, several studies have shown favorable results using values between 10^7^ and 10^9^ CFU of viable microorganisms per dose applied [25,26,27].

International guidelines state that the number of strains declared in probiotic products should be the minimum detected throughout the established shelf life [36]. In the present study, a loss of viability was observed in most of the pharmaceutical formulations tested, which tended to increase over time. Therefore, one strategy to comply with the guidelines would be to use a short shelf life, up to 30 days, for these formulations. Short shelf life is incompatible with industrial processes and continental distribution but is feasible for magistral formulations. Another strategy would be for handlers to estimate the shelf life of the strains used, as was done in this study, and adjust the declared quantities according to the results obtained.

## 5. Conclusions

The results of this study demonstrate that vaginal capsules represent a suitable pharmaceutical form for vaginal use, ensuring the survival of the delivered *Lactobacillus* spp. and contributing to the effectiveness of vulvovaginitis treatment of vulvovaginitis. Capsules made from *Manioht esculenta* polymers are excellent vehicles for maintaining the viability of probiotic *Lactobacillus* strains. These capsules perform better than ointments and ovules and represent an option to improve the treatment of vaginal infections. However, this study presents some limitations. First, we have not studied other aspects of viability besides culturability, such as membrane integrity, which was possibly affected since the probiotic strains were submitted to a stressful process (lyophilization). Further studies are needed to evaluate the viability of these and other strains and the use of different technical excipients, vehicles, or storage instructions that may enhance the activity of probiotics. Moreover, cellular or animal models can help to confirm the efficacy of vaginal capsules, aiming their use as affordable therapeutic tools.

## Figures and Tables

**Figure 1 microorganisms-13-01056-f001:**
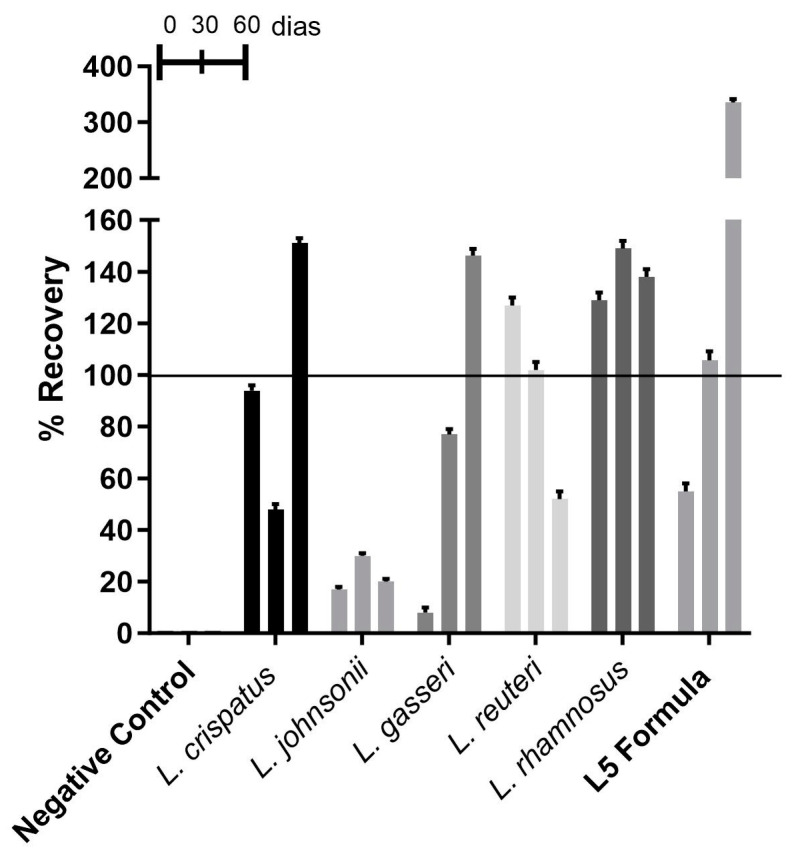
The percentage recovery of *Lactobacillus* spp. from vaginal capsules.

**Figure 2 microorganisms-13-01056-f002:**
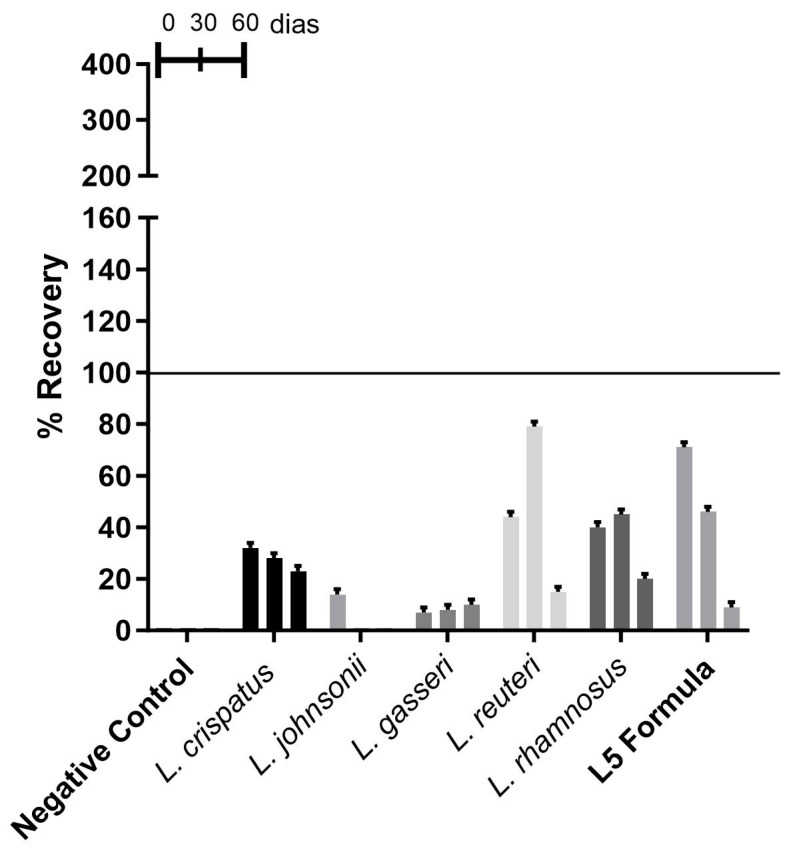
The percentage recovery of *Lactobacillus* spp. from the vaginal ointment formulation.

**Figure 3 microorganisms-13-01056-f003:**
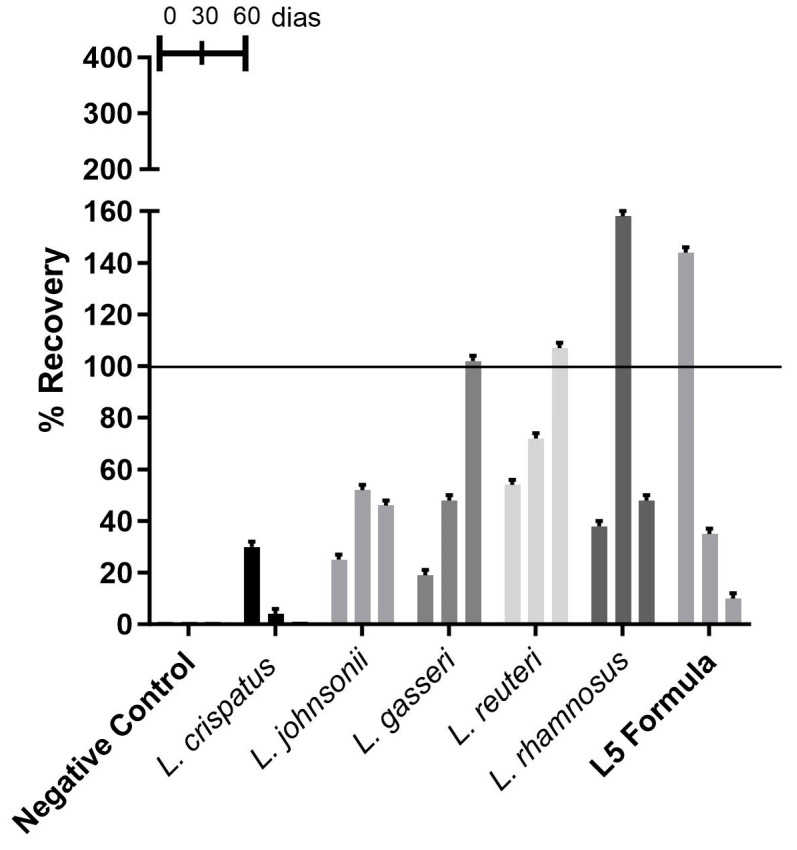
Percentage recovery of *Lactobacillus* spp. from waxy ovule formulation.

**Table 1 microorganisms-13-01056-t001:** Strains used in the present study.

Species	Strain	Lot	Distributor
*Lactobacillus crispatus* *(synonymous with Lactobacillus acidophilus group A2 of Johnson)*	LCr86	Wk20200908004	Active Pharmaceutica
*Lactobacillus johnsonii* *(synonymous with L. acidophilus group B2 of Johnson)*	LJ-G55-81	IK2004	Organic Compounding
*Lactobacillus gasseri*	LG08	WK20201021001	Active Pharmaceutica
*Limosilactobacillus reuteri*	LR-G100	IL2301	Lemma Solutions
*Lacticaseibacillus* *rhamnosus*	LRa05	WK20200321012	Active Pharmaceutica

**Table 2 microorganisms-13-01056-t002:** Composition of vaginal pharmaceutical forms.

Strains of *Lactobacillus* spp.	Quantity, in CFU, of Each Pharmaceutical Form	Composition and Excipients			
		Vaginal Capsule	Gelatinous Ovule	Waxy Ovule	Vaginal Ointment
*L. crispatus*	10^9^ CFU	1 gelatinous capsule lactose s.q. 500 mg	1 ovule s.q. 3 g (gelatin powder 14.3%, potassium sorbate 0.02%, purified water 43%, vegetable glycerin s.q. 100%)	1 ovule s.q. 3g beeswax (*Appis melífera linnaeus*) 5.0% and cacao butter (*Theobroma cacao* 95.0%)	Petrolatum s.q. 1 g
*L. gasseri*	10^9^ CFU				
*L. johnsonii*	10^9^ CFU				
*Limosilactobacillus* *reuteri*	10^9^ CFU				
*Lacticaseibacillus* *rhamnosus*	10^9^ CFU				
L5 formula	5 × 10^9^ CFU				

Note: s.q. (sufficient quantity).

**Table 3 microorganisms-13-01056-t003:** Average weight evaluation of the vaginal capsules.

Vaginal Capsules	Weight		Standard Deviation	Total Weight	Yield Percentage	Average Weight	Results
	Maximum	Minimum					
*L. crispatus*	0.543 g	0.508 g	0.012	0.511 g	103.3%	0.528 g	As per
*L. johnsonii*	0.528 g	0.506 g	0.007	0.528 g	97.7%	0.516 g	As per
*L. gasseri*	0.529 g	0.509 g	0.006	0.519 g	99.9%	0.518 g	As per
*Limosilactobacillus* *reuteri*	0.535 g	0.508 g	0.009	0.517 g	100.0%	0.517 g	As per
*Lacticaseibacillus* *rhamnosus*	0.529 g	0.515 g	0.005	0.516 g	101.4%	0.523 g	As per
L5 formula	0.549 g	0.495 g	0.016	0.549 g	96.8%	0.531 g	As per

**Table 4 microorganisms-13-01056-t004:** Disintegration Times of Pharmaceutical Forms.

Pharmaceutical Form	Average Disintegration Time at 37 °C		
	Gelatinous Ovules	Waxy Ovules	Vaginal Capsules
Strains		Average Time (±SD)	
*L. crispatus*	8 ± 2.0	10.6 ± 0.6	4.3 ± 0.6
*L. johnsonii*	13.6 ± 2.9	11.6 ± 0.6	3.6 ± 0.6
*L. gasseri*	13 ± 1.7	9.6 ± 0.6	4.0 ± 1.0
*Limosilactobacillus* *reuteri*	23.3 ± 0.6	10.6 ± 0.6	4.6 ± 0.6
*Lacticaseibacillus* *rhamnosus*	23.3 ± 0.6	11.3 ± 1.0	3.3 ± 0.6
L5 formula	23.3 ± 1.2	11 ± 1.0	4.3 ± 0.6
Mean	17.5 ± 1.5	10.8 ± 0.7	4.1 ± 0.6

**Table 5 microorganisms-13-01056-t005:** Evaluation of the viability of the formulations studied.

Pharmaceutical Formula	Probiotic	CFU	Real-Time Validity
Vaginal capsules	*L. crispatus*	1 × 10^8^ CFU	60 days
	*L. johnsonii*	1 × 10^8^ CFU	60 days
	*L. gasseri*	1 × 10^7^ CFU	60 days
	*Limosilactobacillus* *reuteri*	5 × 10^8^ CFU	60 days
	*Lacticaseibacillus* *rhamnosus*	1 × 10^9^ CFU	60 days
	L5 formula	5 × 10^8^ CFU	60 days
Vaginal ointment	*L. crispatus*	2 × 10^8^ CFU	60 days
	*L. johnsonii*	0 from 30 days	Infeasible
	*L. gasseri*	1 × 10^8^ CFU	60 days
	*Limosilactobacillus* *reuteri*	1 × 10^8^ CFU	60 days
	*Lacticaseibacillus* *rhamnosus*	2 × 10^8^ CFU	60 days
	L5 formula	1 × 10^8^ CFU	60 days
Waxy ovule	*L. crispatus*	1 × 10^7^ CFU	60 days
	*L. johnsonii*	2 × 10^8^ CFU	60 days
	*L. gasseri*	2 × 10^8^ CFU	60 days
	*Limosilactobacillus* *reuteri*	4 × 10^8^ CFU	60 days
	*Lacticaseibacillus* *rhamnosus*	3 × 10^8^ CFU	60 days
	L5 formula	3 × 10^8^ CFU	60 days

## Data Availability

The raw data supporting the conclusions of this article will be made available by the authors on request.

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
