# Peer review of "Vaginal Capsules: A Viable Alternative for the Delivery of Lactobacillus spp."

_microorganisms, 2025, doi:10.3390/microorganisms13051056_

Round 1
Reviewer 1 Report
Comments and Suggestions for Authors
Review: Vaginal capsules: a viable alternative for the delivery of Lacto- 3 bacillus spp.
The article investigates the stability and viability of four different magistral formulations containing Lactobacillus spp. for potential intravaginal administration to correct dysbiosis. This research is significant because commercial formulations do not ensure viability and therapeutic efficacy of these beneficial bacteria.
The article appears well-structured and comprehensively covers essential aspects of the research.
The literature review is adequate. The authors have justified the objectives.
The laboratory basis of the research component is scientifically correct. However, a microbiologist or pharmacologist with expertise in the research processes should review this section.
While detailed, some sections could benefit from more concise language to improve readability. Reducing redundancy and streamlining the text would enhance clarity.
Including a study design or a diagram is preferable to facilitate understanding.
The limitations and strengths of the study should be mentioned.
The study could lay the foundation for extensive research, with in vivo testing of the formulations to validate their applicability and efficacy.
The article seems well-written and scientifically sound, making it a suitable candidate for publication after minor revision.
Author Response
Dear Editor(s) and reviewer(s):
We greatly appreciate the thorough and thoughtful comments provided on ours submitted article. We made sure that every comment has been carefully addressed and the paper is revised accordingly. Detailed responses to all the reviewer’s comments are below. Please let us know if you still have any questions or concerns about the manuscript. We will be happy to address them all.
Sincerely,
The authors

Reviewer 2 Report
Comments and Suggestions for Authors
The manuscript, entitled “Vaginal capsules: a viable alternative for the delivery of Lactobacillus spp” had described the stability and viability of different strains of Lactobacillus spp. and their recovery percentage from four pharmaceutical forms for vaginal application: vaginal capsules, vaginal ointment, gelatinous ovules, and waxy ovules. It is interesting and important to develop efficient vaginal capsules composed of probiotics.
pH is a critical factor affecting the viability of Lactobacillus spp in vagina. However, the authors failed to discuss its effect on vaginal capsules.
Why the quantity of L5 formula is 5 times of the single strain? Same quantity should be used to compare their efficiency. Since the initial concentration is significantly different, it does not make sense to compare it with the single strain groups. It is obvious 5 times larger in dose is more efficient.
An animal model is suggested to test the efficiency of the formula on vaginal disease treatment and heathy vaginal environmental changes. Also, a blank control without Lactobacillus strains should be included.
Concerning Lactobacillus, new nomenclature had been proposed in 2020 (A taxonomic note on the genus Lactobacillus, DOI 10.1099/ijsem.0.004107) and been well accepted. Please confirm with genus and species are the five strains belong to.
Viability and culturability are different terms. CFU is for culturability not viability. Please refer to DOI 10.1099/mic.0.000786 for viability test methods.
Please double check the formatting. For example, CO2. 1 billion CFU/gram should be transferred to 109 CFU/gram. 500,000 CFU/mL should be transferred to 105 CFU/mL.
Comments on the Quality of English LanguagePlease double check the formatting. For example, CO2. 1 billion CFU/gram should be transferred to 109 CFU/gram. 500,000 CFU/mL should be transferred to 105 CFU/mL.
Author Response

(The authors gave the same response as above.)

Round 2
Reviewer 2 Report
Comments and Suggestions for Authors
The revision is okay.